# Whole exome sequencing reveals novel risk genes of pituitary neuroendocrine tumors

Raitis Peculis[1]*, Vita Rovite[1], Kaspars Megnis[1], Inga Balcere[2], Austra Breiksa[3], Jurijs Nazarovs[3], Janis Stukens[4], Ilze Konrade[1,2], Jelizaveta Sokolovska[5], Valdis Pirags[1,5,6], Janis Klovins[1]

1 Human Genetics and Molecular Medicine, Latvian Biomedical Research and Study Centre, Riga, Latvia, 2 Department of Internal Medicine, Riga Stradins University, Riga, Latvia, 3 Institute of Pathology, Pauls Stradins Clinical University Hospital, Riga, Latvia, 4 Department of Neurosurgery, Pauls Stradins Clinical University Hospital, Riga, Latvia, 5 Faculty of Medicine, University of Latvia, Riga, Latvia, 6 Department of Endocrinology, Pauls Stradins Clinical University Hospital, Riga, Latvia

* raitis@biomed.lu.lv

## Abstract

Somatic genetic alterations in pituitary neuroendocrine tumors (PitNET) tissues have been identified in several studies, but detection of overlapping somatic PitNET candidate genes is rare. We sequenced and by employing multiple data analysis methods studied the exomes of 15 PitNET patients to improve discovery of novel factors involved in PitNET development. PitNET patients were recruited to the study before PitNET removal surgery. For each patient, two samples for DNA extraction were acquired: venous blood and PitNET tissue. Exome sequencing was performed using Illumina NexSeq 500 sequencer and data analyzed using two separate workflows and variant calling algorithms: GATK and Strelka2. A combination of two data analysis pipelines discovered 144 PitNET specific somatic variants (mean = 9.6, range 0–19 per PitNET) of which all were SNVs. Also, we detected previously known *GNAS* PitNET mutation and identified somatic variants in 11 genes, which have contained somatic variants in previous WES and WGS studies of PitNETs. Noteworthy, this is the third study detecting somatic variants in gene *RYR1* in the exomes of PitNETs. In conclusion, we have identified two novel PitNET candidate genes (*AC002519.6* and *AHNAK*) with recurrent somatic variants in our PitNET cohort and found 13 genes overlapping from previous PitNET studies that contain somatic variants. Our study demonstrated that the use of multiple sequencing data analysis pipelines can provide more accurate identification of somatic variants in PitNETs.

## Introduction

Pituitary neuroendocrine tumor (PitNET) or previously often described as clinically significant pituitary adenoma is a rare non-metastasizing endocrine neoplasm affecting approximately 1 person out of 1000 in the general population [1]. Hormone secreting PitNET can cause overproduction of growth hormone (GH), adrenocorticotropic hormone (ACTH), prolactin (PRL) or rarely other hormones leading to various systemic endocrine disorders

**Data Availability Statement:** All sequencing data files are publicly available from The European Genome-phenome Archive (https://ega-archive.org/datasets/EGAD00001006390).

**Funding:** This research was supported by the European Regional Development Fund (ERDF), Measure 1.1.1.1 "Industry-Driven Research" project „Molecular markers of pituitary tumour development, progression and therapy response" (Project No. 1.1.1.1/16/A/066, 2017).

**Competing interests:** The authors have declared that no competing interests exist.

(acromegaly, Cushing's disease, and others). PitNETs without disruption of patient's hormone profile (previously called non-functioning pituitary adenoma) is a subgroup of PitNETs which does not cause disbalance of pituitary hormones. They are usually diagnosed later in life than hormones producing PitNETs and usually due to symptoms related to the effects of the proliferation of PitNET cells on surrounding tissue [2, 3].

High genetic heterogeneity has been observed in pituitary PitNET tissue. Newey *et al.* have sequenced exomes of somatic tissue of seven PitNETs which do not secrete hormones tissues and found 24 potential somatic candidate variants with an average of 3.5 variants per tumor (range from 1 to 7 per neoplasm). They also performed targeted sequencing of potential drivers in *PDGFD*, *NDRG4*, *ZAK* in an independent sample group of 24 tumors, but found no somatic variants in the identified candidates [4]. A study that conducted exome sequencing of 36 somatotroph PitNET tissues discovered 132 somatic variants with average 3 variants per sample ranging from 0 to 13 in the tumor. The *GNAS* mutations were found in 11 tumors (31.4% from 36) and several somatic variants in genes involved in cAMP and calcium signaling were identified [5]. Genome sequencing of 12 somatotroph PitNET have also found mutations in *GNAS*, but no other recurrent somatic variants, and in this report authors demonstrated average 129 somatic variants per tumor genome (range 11–273) and 2.3 variants per exome [6].

Song and colleagues have performed one of the largest studies to date analyzing the genetic landscape of 125 PitNETs [7] identifying *GNAS* and *USP8* mutations repeatedly confirming the role of these genes in PitNET development, as reported by other studies [5, 8–10]. This study also indicated variants in genes *NR3C1*, *KIF5A*, *GRB10* as potential recurrent candidates, and identified *MEN1* somatic variant in two patients without familial MEN syndrome. The authors also found enrichment in somatic variants in Raf/MEK/ERK, PI3K/AKT/mTOR, cAMP, and other signaling pathways [7].

Exome sequencing of eight thyropinomas have not revealed convincing recurrent somatic variants [11] in this study group, but exome analysis of larger group of 42 PitNETs indicated several mutations and insertions/ deletions in genes *ATAD3B*, *BHLHE22*, *KDM2B*, *OR5M1*, *TTN*, *VPS13B* that were encounter in more that one PitNET [12]. No recurrent somatic variants were found in the whole-exome sequencing of 12 prolactinomas [13]. Nemeth et al. have studied mitochondrial variants in 44 PitNET cases and found no significant association with clinical features of the tumor, however, they demonstrated that samples with the highest homoplasmic variant rate had the highest Ki-67 index [14].

Recent study of 134 patients demonstrated novel molecular PitNET classification with deviations from cell lineage guidelines of 2017. High amount of somatic variants per exome (range: 14–247) was reported and recurrence of *USP8* and *GNAS* variants confirmed, as well as presence of chromosomal alterations [15].

Available literature data shows that the number of overlapping candidate genes harboring somatic variants in PitNET like *GNAS* and *USP8* have been confirmed in several independent studies [5, 7, 9, 10], but most of the discovered somatic variants have been detected in the single tumor and have not become novel targets for deeper investigation of PitNET development. Genes that are involved in the development of inherited pituitary PitNET (*AIP*, *MEN1*, *CDKN1B*, *PRKAR1A*) have mostly shown not to contain somatic variants in sporadic cases, with rare exceptions [7, 12]. Additional studies using independent sample groups and larger sample sizes are required to identify overlapping candidate gene to be further investigated for their role in PitNET development mechanisms and as potential targets for improvement of therapy and patient's health care.

In this study, we performed exome sequencing of fifteen PitNETs and used two somatic variant detection algorithms for improved yield and reduced impact of algorithm specific biases to discover somatic variants. We discovered two recurrent somatic variants and two

recurrent genes within our sample as well as 13 genes containing somatic variants which are overlapping with other independent studies. We believe that the sequencing data analysis of PitNET benefits from using multiple variant calling approaches by providing higher confidence in variants that were detected with different methods.

# Materials and methods

## PitNET sample collection

PitNET tissue samples were obtained from transsphenoidal resection leftover material at the Pauls Stradins Clinical University Hospital (Riga, Latvia) from March 2016 to February 2018. Patients with sporadic PitNET were enrolled from 15 consecutive surgeries that provided sufficient amounts of tissue material. Before the surgery PitNET patients were recruited to the government-funded national biobank the Genome Database of Latvian Population (LGDB) and venous blood samples were collected and processed according to the protocol described in [16]. Two written informed consents have been obtained from the patients (1) broad consent for LGDB for use of biological material and medical data for human health and hereditary research, and (2) project-specific consent to the research of pituitary tumors, both studies have been approved by the Central Medical Ethics Committee of Latvia (protocol No. 22.03.07/A7 and 2/18-02-21, respectively). Immediately after surgery PitNET tissue was cut in two parts, (1) was transferred to RNAlater™ Solution (Thermo Fisher Scientific, USA) for DNA/RNA extraction, and (2) was immersed in Dulbecco's Modified Eagle Medium (DMEM) (Thermo Fisher Scientific, USA) containing 1x penicillin/streptomycin solution (GIBCO, USA) for other research activities.

PitNets were classified based on the strongest expressed transcription factor (PIT1, NR5A1 (also known as SF1), TBX18 (also known as TPIT)) and subsequent expression of pituitary hormones (GH, PRL, ACTH, TSHb, LHb, FSHb, CGA). In cases where IHC expression profiles could not be obtained, tumours were classified based on their clinical manifestation.

## Immunohistochemistry of PitNET samples

Immunohistochemical analysis of paraffin-embedded PitNET tissues was performed as an outsourcing service in the Pauls Stradins Clinical University Hospital Institute of Pathology. The following antibodies were used: mouse anti-Growth Hormone (GH) Antibody (MA5-11926), mouse anti-Prolactin (PRL) Monoclonal Antibody (MA5-11998), mouse anti-ACTH Monoclonal Antibody (MA5-13455), mouse anti-Thyroid Stimulating Hormone (TSHb) Antibody (MA5-12159), mouse anti-Luteinizing Hormone (LHb) Monoclonal Antibody (MA5-12138), mouse anti-Follicle Stimulating Hormone (FSHb) Monoclonal Antibody (MA5-12144), mouse anti-Follicle Stimulating Hormone alpha (CGA) Monoclonal Antibody (MA1-82895), rabbit polyclonal NR5a1 antibody (PA5-25030) from Thermo Fisher Scientific, USA, mouse monoclonal Anti-Pit-1 Antibody (G-2) (sc-25258) from Santa Cruz Biotechnology, USA and rabbit polyclonal anti-TBX19 antibody (HPA072686) from Atlas Antibodies, Sweden. The staining was performed on an automated Daco IHC Stain system. Protein expression was evaluated using a 0-3-mark system (1 = <30% positive cells; 2 = 30–70% positive cells; 3 = >70% positive cells). Additional notes were made regarding special characteristics of protein expression patterns (whether cells with positive staining are equally scattered across the sample or arranged in groups; proteins are expressed in the cytoplasm, nucleus, cell membrane or cytoplasmic inclusions).

## DNA extraction

Germline DNA of the patients was isolated from white blood cells according to biobank protocols [16]. Collected PitNET tissue samples were stored in RNAlater™ Solution at +4˚C up to 24 h from surgery, and frozen at -20˚C upon the delivery to the biobank. For DNA/RNA extraction 20–30 mg of tissue samples were lysed using Lysing Matrix D and FastPrep -24 homogenizer (MP Biomedicals, USA). DNA was extracted using AllPrep DNA/RNA Mini Kit (Qiagen, Germany) following the manufacturer's instructions. The concentration of the extracted DNA was measured with Qubit™ dsDNA HS Assay Kit and Qubit® 2.0 Fluorometer (Thermo Fisher Scientific, USA).

## Library preparation and exome sequencing

Exome sequencing was carried out as an outsourcing service to commercial sequencing provider Genera Ltd. (Latvia). DNA libraries were constructed using Illumina Nextera TruSeq Exome kit (Illumina, USA) and sequenced with Illumina NextSeq 500/550 High Output v2 kit (150 cycles) (Illumina, USA) obtaining 75bp paired-end reads on Illumina NexSeq 500 sequencer (Illumina, USA). Exome sequencing was performed in five batches (Table 1).

## Data analysis

Illumina exome target manifest TruSeq Rapid Exome TargetedRegions v1.2 (Illumina, USA) was used to define exome regions in the UCSC hg19 reference. This version of exome has a total length of 45 297 543 bp and includes 214 126 regions.

To detect PitNET somatic variants two algorithms were employed: GATK [17] as implemented in DRAGEN Somatic Pipeline (v3.4.5) and Strelka2 (v2.9.10) [18].

Human genome GRCh37/ hg19 version was used for alignment.

GATK in DRAGEN Somatic Pipeline (v3.4.5) uses a hybrid hardware/software platform to achieve a high speed of sequencing data analysis [19]. The somatic pipeline processes both germline and somatic exome simultaneously and reports variants only present in somatic DNA.

Strelka2 variant caller was run using default parameters. Variant filtering was performed in the same manner across all result files with following parameters: "PASS" in the quality filter, quality score ≥10, and alternative allele has at least four supporting reads. Then germline-somatic variant call file pairs of the same sample were compared and somatic exome unique variants identified with following requirements: germline sample at the somatic variant position has at least 10X coverage, somatic variant is not located within or adjacent to (one nucleotide distance) homopolymer (≥4 bases) sequence and variant is not located in a repeated sequence region. Somatic variants colocated with existing database SNPs were registered and

**Table 1. Batch information of PitNET exome sequencing.**

| Batch # (date) | Germline DNA samples | Somatic DNA samples |
|---|---|---|
| 1 (jul 2017) | PN02 | PN01 |
| 2 (oct 2017) | PN01; PN03; PN04; PN05; PN06; PN07 | PN02; PN03; PN04; PN05; PN06; PN07 |
| 3 (nov 2017) | PN08; PN09; PN10; PN11; PN12 | PN08; PN09; PN10; PN11 |
| 4 (jul 2018) | PN13; PN14; PN15 | PN12; PN13; PN14; PN15 |
| 5 (jan 2019) | PN15 | PN15 |

The sequencing data has been deposited to The European Genome-phenome Archive with access ID EGAC00001001730.

filtered if the population MAF above 0.1%. Overlapping variants between both workflows were registered as true positives. Variant Effect Predictor as implemented in ensembl.org was used to predict functional consequences of variants [20]. Copy number analysis was performed using CoNIFER [21] and XHMM [22]. Variant frequency data from gnomAD v2.1.1 database was used to present results [23].

# Results

## Characteristics of PitNET samples

We obtained whole-exome sequencing data of paired white blood cells and PitNET tissue from 15 patients. Mean depth of coverage was 50.3X (range 15.9–82.2X, SD = 17.2). Individual PitNET patient information is shown in Fig 1. Clinically nine PitNETs were defined as NFPA, three were GH secreting and two PRL secreting tumors. Immunohistochemistry analyses identified that PIT1 cell lineage marker dominates in four PitNETs, gonadotroph PitNET marker NR5A1 (SF1) is the most abundant in five tumours and corticotroph cell lineage marker TBX19 (TPIT) has highest expression in one PitNET (Fig 1 and S1 File). All PitNETs were larger than 10 mm in at least one dimension in MRI data measurement and 60% of patients were men.

## Immunohistochemistry staining results of PitNET samples

Immunohistochemistry data of paraffin-embedded PitNET tissues were obtained for 11 PitNET patients using hormone antibodies (except CGA, where eight samples were analysed) and also 11 but different samples for lineage marker antibodies (S1 File). Concordance of hormone antibody immunohistochemistry with clinical diagnosis was observed in eight out of 11 patients, meanwhile nine out of 11 patients had agreement between clinical diagnosis and ICH data for PitNET lineage markers (PIT1, NR5A, TBX19). Most of NFPA clinical diagnoses actually were gonadotroph hormones expressing PitNETs. Few notable exceptions were observed: PN02 (with clinical NFPA diagnosis) was positive for GH and ACTH as antibody staining showed up to 30% of cells containing these hormones. PN04 (NFPA) had no detectable hormone expression, but PN05 (NFPA) was expressing every tested hormone, with high levels of GH, PRL and ACTH showing up in the data, although the highest expression level was for glycoprotein alpha subunit (required to produce gonadotropic hormones). PN10 (clinically PRL secreting PitNET) and PN14 (clinically GH secreting PitNET) were also expressing high levels of the glycoprotein alpha subunit.

PitNET cell lineage markers showed difference from clinical diagnosis in PN02 and PN10. PN02 is expressing corticotroph cell lineage marker in 30–70% of its cells. PN10 which clinically manifests as prolactin secreting PitNET is expressing higher amount of NR5A which is gonadotroph cell lineage marker than PIT1 associated with development of prolactin secreting PitNET.

## Identification of somatic exome variants

Results of exome sequencing of 15 PitNETs show a total of 144 PitNET specific somatic variants (mean = 9.6, range 0–31 per PitNET) of which all were SNVs. We employed two somatic variant search algorithms (GATK and Strelka2) and reported only somatic variants where both of the employed algorithms agreed.

Most of the discovered variants (N = 91) are located within the exons of genes and 58 of those are changing the amino acid sequence of the protein (range per tumor 0–11, mean 3.9) while 22 are synonymous SNVs, rest being 3 prime UTR variants (N = 4), 5 prime UTR

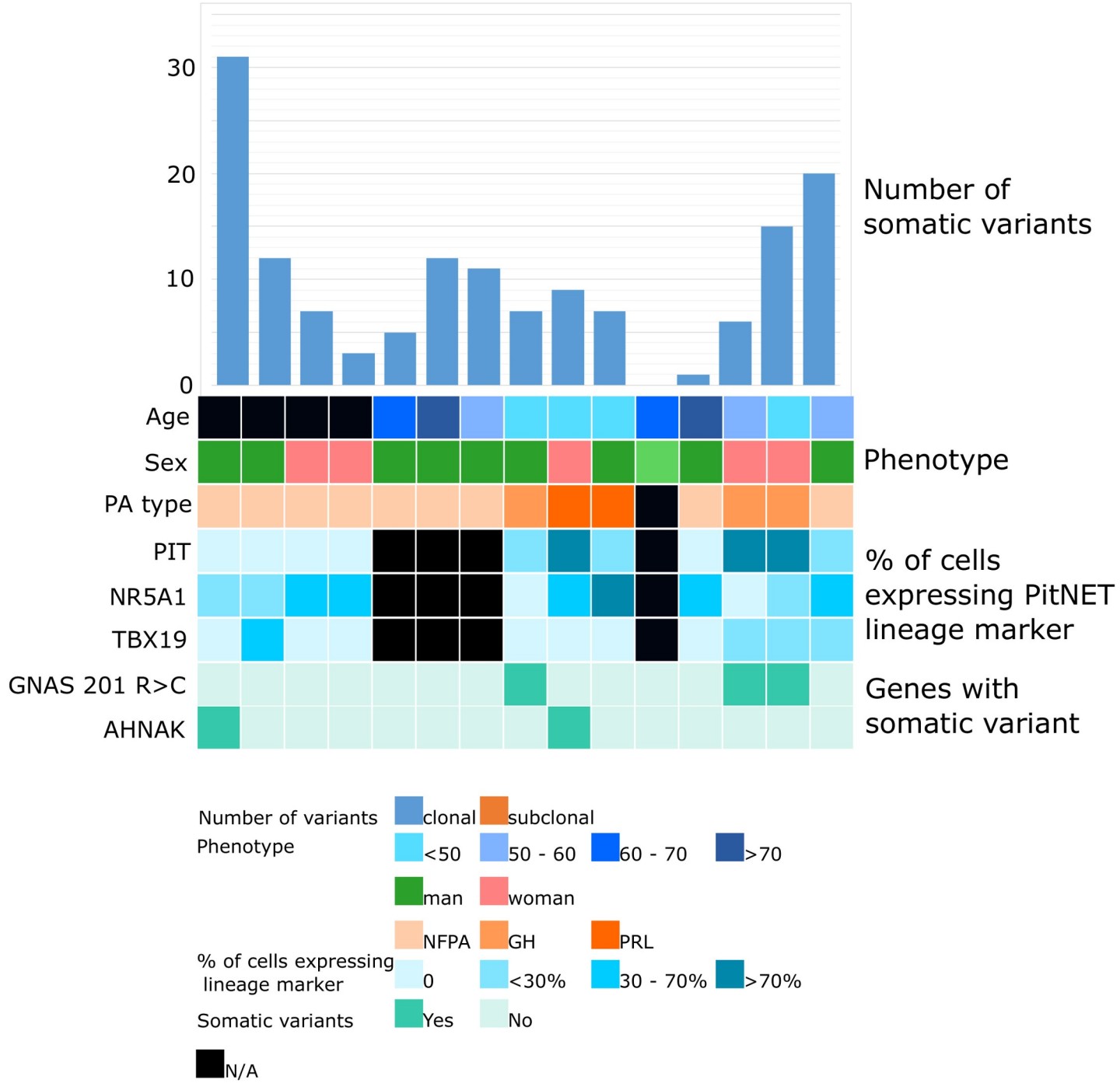

**Fig 1. Amount of somatic variants, clinical information of PitNET patients (PitNET clinical type, age at diagnosis, sex), % of cells expressing lineage markers and genes with recurrent somatic variants are shown for all 15 PitNET patients.** PitNETs are listed in ascending order from left to right.

variants (N = 3) and non coding transcript exon variant (N = 4).The most common location of variants outside the coding sequence were introns (N = 44). The rest of the variants were located in untranslated gene regions, non-coding transcripts, predicted splice sites and outside genes altogether (S1 Table).

Assessment of possible functional consequences of SNVs identified eight SNVs from seven PitNETs which were rated within the top 0.1% deleterious SNVs in the human genome according to Combined Annotation Dependent Depletion (CADD) matric (CADD Phred score >30) and further 34 SNVs within the top 1% deleterious SNVs in the human genome (Phred score >20). All but two of the analyzed PitNET samples contain at least one highly deleterious SNV (Phred score >20) (S1 Table). Three PitNETs have a somatic previously known PitNET associated *GNAS* mutation (position R844C in transcript ENST00000371100.4). Three genes (*GNAS*, *AHNAK* and *PAML2/AKAP2*) have a mutation in at least two PitNETs but in different positions. While transcribed RNA pseudogene AC002519.6 contains the same genetic variant in the same position in two PitNETs (Tables 2 and 3). Additionally, we identified 13 overlapping genes with other PitNET and neoplasm genetic studies (*GNAS* [8], *CRTC3*, *SMARCA4* [24], *KIT* [25], *PLA2G6*, *TESK1* [5], *TSC2* [4], *KLHL4*, *RYR1* [6], *POLR3B*, *KHDRBS2*, *LAMA1*, *TUBGCP6* [7] and *AHNAK* [15]).

Copy number analysis was performed using CoNIFER [21] and XHMM [22] algorithms did not reveal presence of copy number alterations in analysed samples.

## Discussion

This is the first study demonstrating the advantages of exome sequencing analysis using multiple data analysis workflows in the research of PitNET genetic composition. Despite the existing knowledge that in the exome and genome sequencing variant calling results from the single pipeline are incomplete [19, 26], no current literature data is examining the consequences of this for the discovery of PitNET somatic variants. It has been demonstrated that non-overlapping variants from different pipelines mostly are true positive nucleotide changes [19] yet most raw non-overlapping calls of both GATK and Strelka2 in our data showed low alternate allele frequency and other variant calling issues. Therefore, our results provide a more comprehensive approach to identify the genetic composition of PitNET. Exome analysis of blood–tumor pairs from 15 PitNET patients using two distinct data analysis workflow showed increased mean number of somatic variants per PitNET (9.6) compared to previous publications (range 2.3–3.5) [4, 6, 7] and is supported by two of the latest reports [12, 15] showing number of somatic variants often above 10 per tumor. Such an improved approach can lead to the identification of novel mechanisms of PitNET development. Although studies published in literature already encompass several hundred exomes and genomes, recurrent somatic variants beyond established risk factors (*GNAS*, *USP8*) are virtually non-existent in sporadic PitNET cases. This could be explained that PitNETs form according to Knudson's two hit hypothesis [27] and large amounts of nucleotide change combinations can result in a PitNET. Also it is possible that constraints of current exome and genome sequencing technology and analysis algorithms are unable to find certain amounts of somatic variants, therefore important

**Table 2. Somatic variants detected in more than one PitNET sample.**

| CHR:pos | Ref/ Alt | PitNET samples | Alt allele fraction | Gene | Consequence | CADD Phred | gnomAD freq |
|---|---|---|---|---|---|---|---|
| chr20:57484420 | C/T | PN14 | *41.7%* | *GNAS* | Missense variant | 34 | 0 |
| | | PN13 | *41.0%* | | | | |
| | | PN08 | *45.4%* | | | | |
| chr16: 31818244 | C/G | PN01 | *10.3%* | *AC002519.6* | Downstream gene variant, rs111534922 | 2.24 | 0 |
| | | PN07 | *16.7%* | | | | |

Chr–chromosome, pos–position, PitNET–pituitaryneuroendocrine tumor, ref–reference allele, alt–alternative allele, CADD–CombinedAnnotation Dependent Depletion, freq–frequency.

**Table 3. Somatic variants found in the same gene and in more than one PitNET sample.**

| Chr | Position | PitNET sample | Ref/Alt | Gene | Consequence | CADD Phred | gnomAD freq |
|-----|----------|---------------|---------|------|-------------|------------|-------------|
| chr11 | 62283636 | PN02 | T/C | AHNAK | 3 prime UTR variant | 0.57 | 0 |
| chr11 | 62289022 | PN01 | T/A | | synonymous variant | 0.008 | 0.003% |
| Chr9 | 112625173 | PN06 | G/T | PALM2/AKAP2 | Intron variant | 1.26 | 0 |
| Chr9 | 112687347 | PN15 | T/A | | Missense variant, COSV57111433 | 19.12 | 0 |

Chr–chromosome, PitNET–pituitary neuroendocrine tumor, ref–reference allele, alt–alternative allele, CADD–Combined Annotation Dependent Depletion.

findings are missing from literature. The third reason for low overlap in results of the published PitNET genomic studies could be bias introduced by sampling as surgery materials are obtained mostly from patients with PitNETs which are unresponsive to drug treatment.

Regarding IHC results confirmed that most of the PitNET presented clinical phenotype according to their expressed lineage markers and hormones. Only one of the IHC tested clinically non-functional PitNET (PN02) differed from the rest by having TPIT cell lineage marker ekspressed in 30%– 70% of observed cells rather than *NR5A1*. On the other hand, PN10 was presenting hyperprolactinemia phenotype, but the highest proportion of cells (>70%) was expressing *CGA* and *NR5A1* indicating gonadotroph PitNET [28], nevertheless, PN10 was also expressing *POU1F1* cell lineage marker and PRL in up to 30% of cells, probably causing more noticeable hyperprolactinemia induced clinical phenotype. Also we observed that four out of five PRL or GH secreting PitNETs were expressing more than one cell lineage marker while the same was true in two out of six PitNETs which were designated as non-functioning PitNET. Similar differences between secretion subtype and cell lineage markers has been observed in other studies [15] and they are of low concern regarding detection of somatic tumor variants.

We have, for the first time, detected recurrent somatic variant located in exactly the same position in an *AC002519.6* of the two PitNETs (PN01 and PN07) (Table 2). The variant is listed in dbSNP with identifier rs111534922, but has 0 MAF across all populations. As a transcribed RNA pseudogene, there is little information about it in literature [29].

Total transition to transversion ratio was 1.3. This is in line with literature data where variability of transition/transversion ratio across PitNET is high, although the majority have it between 1.2 and 3 [6].

It is fairly common to observe somatic mutations in *GNAS*, *USP8* and *MEN1* when performing WES or WGS in PitNET patients [6, 7, 10]. We also detected common *GNAS* somatic mutation at chr20:57484420, rs11554273 (also known as *GNAS* 201 R>C) in three PitNET patients (all expressing PIT) (Table 3). *GNAS* rs11554273 has virtually zero frequency of a minor allele in the general population sample, but is repeatedly found in PitNET tissues [6, 7] and it is heterogeneously impacting clinical properties of somatotroph PitNET [30].

The only other recurrent gene with somatic variants in more than one tumor was *AHNAK* (PN01 and PN02), but impact on gene function for both variants is hard to pinpoint due their supposedly benign consequence (synonymous variant and 3' untranslated region variant) and low predicted consequence phred score by CADD (0.009 and 0.59 respectively). Interesting that somatic variants in *AHNAK* have been reported before in PitNET [15]. *AHNAK* previously has been implicated in both pro-tumorigenic [31–33] and anti-tumor [34] roles of which the most interesting in PitNETs would be its involvement in epithelial-mesenchymal transition [35]. Nevertheless, its role in PitNETs should be defined in additional, more focused studies. Another way to perform the research of somatic variants of PitNET is to evaluate their presence in the analyzed sample as an allelic fraction. Although limitations of this approach are

clear (random distribution of allelic fraction, higher false discovery rate, allele-specific library amplification and PitNET tissue sample pollution with non-tumor cells), we speculate that at least part of somatic variants with allelic fraction below 20% (designated as "subclonal") arose as a result of increased rate of cell division and diminished capability to repair DNA replication errors in an already established PitNET. This may indicate that the majority of true subclonal variants are present only in part of PitNET cells and promotes a question if these genetically heterogeneous cell groups have an impact on overall PitNET properties and clinical characteristics. Fifty-eight (40.35%) of detected somatic variants in this study were subclonal, showing that multiple variation detection algorithms identify such variants and that they are common in PitNETs.

Several genes that have been reported with somatic variants (without dbSNP identifier and gnomAD frequency) in previous PitNET exome and genome sequencing studies, also harbor somatic variants in tumors of our PitNET patients. These genes are (*GNAS* [8], *TSC2* [4], *KLHL4* [6], *POLR3B*, *KHDRBS2*, *LAMA1*, *TUBGCP6* [7], *RYR1* [7, 6], *PLA2G6*, *TESK1* [5], *AHNAK* [15] but their somatic variants are all novel and in all cases distant enough to exclude variant hotspots or disruption of certain part of the protein. These genes could be potential candidates for further studies as functional pathways impacted by these genes lead to the formation of PitNETs. *RYR1* has been reported in previous PitNET studies at least twice [6, 7] and had not been highlighted as PitNET associated. *RYR1* is not expressed in rat pituitary [36], no data about this gene expression in human pituitary have been reported, if indeed *RYR1* is related to PitNET development this could also be due to an indirect effect through gene-gene interaction or linkage disequilibrium with another causative factor. *RYR1* functions as a regulator of inner cell $Ca^{2+}$ flow, the dysfunctions of this factor have been implicated in heart conditions, myopathies and neurodegenerative disease [37]. It has been shown that somatic variation in calcium regulation related genes are common in PitNETs [5], therefore this is an interesting target for potential functional studies in PitNET development.

*LAMA1* [38, 39] and *AHNAK* [35] has been implicated in tumor metastasis while *TSC2* is considered tumor suppressor (reviewed in [40]). Relation of other overlapping genes with PitNET is more obscure.

Tumor-related gene *KIT* [25], was also detected to have a somatic variant. (synonymous and splice site variant) was found in PN02. This somatic variant was estimated to be relatively benign.

Several limitations can be identified for our study. The deviation of PitNET type distribution in our sample from general distribution of clinical PitNET patient type in our population and literature [41] could be explained with relatively small sample size and/ or overrepresentation of large PitNET (over 10 mm in largest dimension)which are more likely to be treated with surgery and therefore providing tumor cells for genetic research. The unusual gender disbalance also is most likely attributed to chance as the material from surgeries from March 2016 to February 2018 were collected. And finally, chromosomal aberrations in PitNET are often investigated by using array genotyping which could add information on copy number variations across PitNET genomes. We did not detect high confidence copy number variations in our samples confirming problems in such analysis from exome data mentioned in M. Fromer and S. M. Purcell, 2015 [22].

In conclusion, besides GNAS and AHNAK, we have identified PitNET 11 more genes overlapping from previous PitNET studies which contain somatic variants of which several (RYR1, LAMA1, AHNAK, TSC2) could be highlighted as potential candidates for further PitNET research. The role of these genes and somatic variants in PitNET development should be investigated in further studies. We propose the use of multiple sequencing data analysis pipelines as they can provide more accurate identification of somatic variants in PitNETs.

## Supporting information

**S1 Table. Description of all somatic variants of PitNET exomes.**
(XLSX)

**S1 File. Example photographs of immunohistochemistry detection of PIT1 cell lineage marker, corticotroph cell lineage marker TBX19 (TPIT) and gonadotroph PitNET cell lineage marker NR5A1 (SF1).**
(ZIP)

## Acknowledgments

The authors acknowledge the Latvian Biomedical Research and Study Centre and the Genome Database of the Latvian Population for providing infrastructure, biological material and data.

## Author Contributions

**Conceptualization:** Raitis Peculis, Inga Balcere, Ilze Konrade, Jelizaveta Sokolovska, Valdis Pirags, Janis Klovins.

**Data curation:** Inga Balcere, Austra Breiksa, Jurijs Nazarovs, Janis Stukens, Jelizaveta Sokolovska.

**Formal analysis:** Raitis Peculis, Austra Breiksa.

**Funding acquisition:** Janis Klovins.

**Investigation:** Kaspars Megnis, Jurijs Nazarovs, Ilze Konrade, Valdis Pirags.

**Project administration:** Janis Klovins.

**Supervision:** Vita Rovite, Janis Klovins.

**Validation:** Vita Rovite, Janis Klovins.

**Writing – original draft:** Raitis Peculis, Inga Balcere, Austra Breiksa, Jurijs Nazarovs, Ilze Konrade.

**Writing – review & editing:** Raitis Peculis, Vita Rovite, Kaspars Megnis, Inga Balcere, Austra Breiksa, Janis Stukens, Ilze Konrade, Jelizaveta Sokolovska, Valdis Pirags, Janis Klovins.

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
