## [Decision Letter · Decision Letter 0]

29 Jun 2021

PONE-D-20-40370

Whole exome sequencing reveals novel risk genes of pituitary neuroendocrine tumors

PLOS ONE

Dear Dr. Peculis,

Thank you for submitting your manuscript to PLOS ONE. After careful consideration, we feel that it has merit but does not fully meet PLOS ONE’s publication criteria as it currently stands. Therefore, we invite you to submit a revised version of the manuscript that addresses all the points that have been raised by the three reviewers during the review process (see below).

We look forward to receiving your revised manuscript.

Kind regards,

Raul M. Luque, PhD

Academic Editor

PLOS ONE

Journal Requirements:

The authors acknowledge the Latvian Biomedical Research and Study Centre and the Genome Database of the Latvian Population for providing infrastructure, biological material and data.

This research was supported by the European Regional Development Fund (ERDF), Measure 1.1.1.1 “Industry-Driven Research” project „Molecular markers of pituitary tumour development, progression and therapy response” (Project No. 1.1.1.1/16/A/066, 2017).

This research was supported by the European Regional Development Fund (ERDF), Measure 1.1.1.1 “Industry-Driven Research” project „Molecular markers of pituitary tumour development, progression and therapy response” (Project No. 1.1.1.1/16/A/066, 2017).

4. We note that you are reporting an analysis of a microarray, next-generation sequencing, or deep sequencing data set. PLOS requires that authors comply with field-specific standards for preparation, recording, and deposition of data in repositories appropriate to their field. Please upload these data to a stable, public repository (such as ArrayExpress, Gene Expression Omnibus (GEO), DNA Data Bank of Japan (DDBJ), NCBI GenBank, NCBI Sequence Read Archive, or EMBL Nucleotide Sequence Database (ENA)). In your revised cover letter, please provide the relevant accession numbers that may be used to access these data. For a full list of recommended repositories, see http://journals.plos.org/plosone/s/data-availability#loc-omics or http://journals.plos.org/plosone/s/data-availability#loc-sequencing. 

Reviewers' comments:

Reviewer's Responses to Questions

**Comments to the Author**

1. Is the manuscript technically sound, and do the data support the conclusions?

Reviewer #1: Partly

Reviewer #2: Yes

Reviewer #3: Partly

2. Has the statistical analysis been performed appropriately and rigorously? 

Reviewer #1: N/A

Reviewer #2: Yes

Reviewer #3: I Don't Know

3. Have the authors made all data underlying the findings in their manuscript fully available?

Reviewer #1: Yes

Reviewer #2: Yes

Reviewer #3: Yes

4. Is the manuscript presented in an intelligible fashion and written in standard English?

Reviewer #1: Yes

Reviewer #2: Yes

Reviewer #3: Yes

5. Review Comments to the Author

Reviewer #1: In this manuscript, the authors described their findings of risk genes from patients with pituitary neuroendocrine tumors with whole exome sequencing data. The authors have improved their analysis methods and changed the manuscripts according to the suggestions from previous reviewers. This work could be helpful for future research and treatment of this disease. Thus, I suggest accepting this manuscript if the authors can go through the manuscript carefully and made improvements seriously.

1. For methods, include the details about how genes and variants were annotated. Which version of gnomAD data is used? Some brief explanation about columns in table S1 is necessary.

2. Improve the language. “Is the manuscript presented in an intelligible fashion and written in standard English?” The authors replied that “Thank you for this comment we thoroughlychecked the manuscript for…” It is unexpected that “thoroughlychecked” should be “thoroughly checked”.

I will list some examples:

In abstract: “We sequenced and employing multiple data analysis methods studied the exomes of 15 PitNET patients to improve discovery of novel factors involved in PitNET development.”

Line 96: “Extended studies using independent sample groups and larger sample sizes are required to reveal overlapping candidates to be further investigated for their role in PitNET development mechanisms and potential improvement of therapy and patient's health care.” Hard to understand.

Line 102: “We discovered two recurrent somatic variants within our sample and two recurrent genes within our sample as well as 13 genes compared to other independent studies.” It should be “our samples”? It is not necessary to use “our sample” twice. “samples”?

Line 299: “but both variants are unlikely to impact gene function both due their consequence (synonymous variant and 3’ untranslated region variant) and consequence prediction phred score by CADD (0.009 and 0.59 respectively).”

Line 354: “In conclusion we have identified PitNET 11 more genes overlapping from previous PitNET studies which contain somatic variants of which several (RYR1, LAMA1, AHNAK, TSC2) could be highlighted as potential candidates for further PitNET research.” Line 36: “In conclusion, we have identified two novel PitNET candidate genes (AC002519.6 and AHNAK) with recurrent somatic variants in our PitNET cohort and found 13 more genes overlapping from previous PitNET studies that contain somatic variants.” The description seems wrong. 11 or 13? “more genes overlapping from previous studies”? My understanding is overlapping means recurrent detection of these genes, and it is hard to know what “more” means here.

3. Please use a table to describe the sequenced runs for each sample. Current description in “Library preparation and exome sequencing” include partial details. If there is no batch effect, the authors may simplify the descriptions. Current description only included partial details about how samples were sequenced.

4. “Most of the discovered variants (N = 93) are located within the coding sequence of genes and 58 of those are changing the amino acid sequence of the protein (range per tumor 0 – 11 , mean 3.9) while 22 are synonymous SNVs.” 58 + 22 = 80. What are the rest 13 variants?

5. No summary for “Immunohistochemistry of PitNET samples”.

6. Line 104: “We show that the sequencing data analysis of PitNET benefits from using multiple variant calling approaches”. Past tense should be used. Also, current evidence does not support the conclusion that “multiple variant calling approaches” benefit “data analysis of PitNET”.

Minor:

Line 27: change “Two samples” to “For each patient, two samples”.

Line 345: change “largePitNET” to “large PitNET”?

Reviewer #2: The manuscript PONE-D-20-40370 entitled ‘Whole exome sequencing reveals novel risk genes of pituitary neuroendocrine tumors’ identify novel mutations in pituitary tumors. The study is really interesting from the clinical point of view, since it validates some of previously described mutations and provides evidence of new candidate genes. It is not a large cohort, but it is still informative work. Nevertheless, there are some issues that may be addressed.

1. The pituitary tumor subtype is a key element of these kind of studies, and it must be indicated in the abstract and introduction. Although it may be clear in the figure, a more detailed characterization including IHC should be included, perhaps a non-gonadotroph but ACTH- or GH-silent adenomas were included, which may be interesting.

2. Since there is a vast range of mutation per sample, from 0 to 31, is there any correlation with clinical or molecular characteristics?

3. In the discussion, the authors indicate that the use of different workflows is advantageous, nevertheless, they do not really compare the methods, since they combine the results of the different workflows used.

Reviewer #3: This article reported results for the identification of two novel PitNET candidate genes ( AC002519.6 and AHNAK) with recurrent somatic variants in a PitNET cohort and found 13 more genes overlapping from previous PitNET studies containing somatic variants. This article does not show relevant and robust results in the field of PitNETs. Thus, there are major aspects that could be review aiming to improve the manuscript:

- The authors should considerably improve the result section. In general, the results are insufficient since the authors could acquire more results from the methodology used and data analysis. Then, we encouraged the author to deeply analyzed the data obtained from both methodologies used (GATK and Strelka2).

- The sample cohort is only for 15 samples, being only three GH secreting and two PRL secreting tumors. Thus, the authors should complete the cohort with more samples for each subtype to avoid bias in the data analysis or reanalyzed other cohorts with the same algorithm in order to corroborate the data of your cohort.

As a minor comment, I would like to suggest that the expression level in figure 1 should be represented as a log2(FC) and not as a percentage because they could be ambiguous.

In conclusion, this article needs to improve major/minor aspects to achieve an appropriate relevance.

6. PLOS authors have the option to publish the peer review history of their article (what does this mean?). If published, this will include your full peer review and any attached files.

Reviewer #1: No

Reviewer #2: No

Reviewer #3: **Yes: **Antonio C. Fuentes-Fayos

---

## [Author Response · Author response to Decision Letter 0]

4 Jan 2022

Response to Reviewers

We thank editor and reviewers for considering this work and involvement to improve quality, relevance and readability of this research article. We looked into comments made by reviewers and did our best to improve the manuscript according to reviewers' suggestions.

Specific answers to all the issues raised by reviewers are listed below:

1. Is the manuscript technically sound, and do the data support the conclusions?

Reviewer #1: Partly

Reviewer #2: Yes

Reviewer #3: Partly

Author answer: We have changed multiple instances in the manuscript according to further reviewer comments including additional information on immunohistochemistry results, sequencing batches, improved data and legend if Fig 1, added legend to S1 and expanded discussion. Also we aligned conclusions more closely to the data.

2. Has the statistical analysis been performed appropriately and rigorously?

Reviewer #1: N/A

Reviewer #2: Yes

Reviewer #3: I Don't Know

3. Have the authors made all data underlying the findings in their manuscript fully available?

Reviewer #1: Yes

Reviewer #2: Yes

Reviewer #3: Yes

4. Is the manuscript presented in an intelligible fashion and written in standard English?

Reviewer #1: Yes

Reviewer #2: Yes

Reviewer #3: Yes

Author answer: We thank for careful check of the manuscript requirements

 

Reviewer #1: In this manuscript, the authors described their findings of risk genes from patients with pituitary neuroendocrine tumors with whole exome sequencing data. The authors have improved their analysis methods and changed the manuscripts according to the suggestions from previous reviewers. This work could be helpful for future research and treatment of this disease. Thus, I suggest accepting this manuscript if the authors can go through the manuscript carefully and made improvements seriously.

1. For methods, include the details about how genes and variants were annotated. Which version of gnomAD data is used? Some brief explanation about columns in table S1 is necessary.

Author answer: Thank you for your time and effort to help us improve the manuscript! We used gnomAD v2.1.1 for variant frequency data. We have included this information in the manuscript line 224: "Variant frequency data from gnomAD v2.1.1 database was used to present results." We have added additional tab "Legend" to the S1 table, explaining all the column names and abbreviations within them.

2. Improve the language. “Is the manuscript presented in an intelligible fashion and written in standard English?” The authors replied that “Thank you for this comment we thoroughlychecked the manuscript for…” It is unexpected that “thoroughlychecked” should be “thoroughly checked”.

Author answer: Thank you for careful reviewing. This is unfortunate technical issue of dropping "space" between words is caused by moving manuscript between authors who are using different versions of MS Office. It is caused during the saving procedure, therefore, impossible to prevent if using affected version. We will correct the spelling mistakes and use newest MSOffice version to ensure these mistakes are not repeated.

I will list some examples:

In abstract: “We sequenced and employing multiple data analysis methods studied the exomes of 15 PitNET patients to improve discovery of novel factors involved in PitNET development.”

 Author answer: Changed to: “We sequenced and by employing multiple data analysis methods studied the exomes of 15 PitNET patients to improve discovery of novel factors involved in PitNET development.”

Line 96: “Extended studies using independent sample groups and larger sample sizes are required to reveal overlapping candidates to be further investigated for their role in PitNET development mechanisms and potential improvement of therapy and patient's health care.” Hard to understand.

Author answer: We rephrased the sentence to convey our message clearer: "Additional studies using independent sample groups and larger sample sizes are required to identify overlapping candidate gene to be further investigated for their role in PitNET development mechanisms and as potential targets for improvement of therapy and patient's health care."

Line 102: “We discovered two recurrent somatic variants within our sample and two recurrent genes within our sample as well as 13 genes compared to other independent studies.” It should be “our samples”? It is not necessary to use “our sample” twice. “samples”?

Author answer: We agree to the reviewer and changed sentence to: “We discovered two recurrent somatic variants and two recurrent genes within our sample as well as 13 genes containing somatic variants which are overlapping with other independent studies.”

Line 299: “but both variants are unlikely to impact gene function both due their consequence (synonymous variant and 3’ untranslated region variant) and consequence prediction phred score by CADD (0.009 and 0.59 respectively).”

Author answer: Indeed, this part of the sentence is confusing. We have edit that to: "but impact on gene function for both variants is hard to pinpoint due their supposedly benign consequence (synonymous variant and 3’ untranslated region variant) and low predicted consequence phred score by CADD (0.009 and 0.59 respectively)."

Line 354: “In conclusion we have identified PitNET 11 more genes overlapping from previous PitNET studies which contain somatic variants of which several (RYR1, LAMA1, AHNAK, TSC2) could be highlighted as potential candidates for further PitNET research.” Line 36: “In conclusion, we have identified two novel PitNET candidate genes (AC002519.6 and AHNAK) with recurrent somatic variants in our PitNET cohort and found 13 more genes overlapping from previous PitNET studies that contain somatic variants.” The description seems wrong. 11 or 13? “more genes overlapping from previous studies”? My understanding is overlapping means recurrent detection of these genes, and it is hard to know what “more” means here.

Author answer: The confusion probably arises because we have incorrectly used "more" in both places. In discussion we already extended description about GNAS and AHNAK and then there are "11 more" (additional) overlapping genes (11+2=13) as you correctly deduced. We have dropped "more" from line 38 [36] and edited line 391 [354] to: "In conclusion, besides GNAS and AHNAK, we have identified PitNET 11 more genes overlapping from previous PitNET studies which contain somatic variants of which several (RYR1, LAMA1, AHNAK, TSC2) could be highlighted as potential candidates for further PitNET research."

3. Please use a table to describe the sequenced runs for each sample. Current description in “Library preparation and exome sequencing” include partial details. If there is no batch effect, the authors may simplify the descriptions. Current description only included partial details about how samples were sequenced.

Author answer: As suggested by reviewer we have introduced additional table in section "Library preparation and exome sequencing" and edited text starting from line 167: "Exome sequencing was performed in five batches (Table 1)." Subsequently, table numbering was adjusted in the rest of manuscript.

4. “Most of the discovered variants (N = 93) are located within the coding sequence of genes and 58 of those are changing the amino acid sequence of the protein (range per tumor 0 – 11 , mean 3.9) while 22 are synonymous SNVs.” 58 + 22 = 80. What are the rest 13 variants?

Author answer: Thank you for pointing out this issue. Counting of somatic variants in our study is indeed so not clear and two different variant properties contribute to that: 1) in two cases somatic variants are located on the genomic position which belongs to two genes located on forward and reverse strands, therefore consequences are more than somatic variants and this is compounded by the fact that two variants are found in more than one sample making unique somatic variant count even more different; 2) there are somatic variants with multiple predicted consequences: five variants with two, one variant with three (S1 "Consequence" column). Taken together this causes challenging environment for counting variant consequences and describing them in clear, unambiguous way. To answer the question about rest 13 variants: there are four 3_prime_UTR_variants, three 5_prime_UTR_variants, four non_coding_transcript_exon_variants (S1), total 11 and the two missing are incorrectly counted double consequence variants: missense_variant, splice_region_variant. To make this sentence clearer we have edited this sentence and corrected the error counting double consequence variants twice: "Most of the discovered variants (N = 91) are located within the exons of genes and 58 of those are changing the amino acid sequence of the protein (range per tumor 0 – 11, mean 3.9) while 22 are synonymous SNVs, rest being 3 prime UTR variants (N = 4), 5 prime UTR variants (N = 3) and non-coding transcript exon variant (N = 4)."

5. No summary for “Immunohistochemistry of PitNET samples”

Author answer: Thank you for pointing this out, we have added more elaborate description in "Discussion" section about immunohistochemistry results: "Regarding IHC results confirmed that most of the PitNET presented clinical phenotype according to their expressed lineage markers and hormones. Only one of the IHC tested clinically non-functional PitNET (PN02) differed from the rest by having TPIT cell lineage marker ekspressed in 30% – 70 % of observed cells rather than NR5A1. On the other hand, PN10 was presenting hyperprolactinemia phenotype, but the highest proportion of cells (>70%) was expressing CGA and NR5A1 indicating gonadotroph PitNET [28], nevertheless, PN10 was also expressing POU1F1 cell lineage marker and PRL in up to 30% of cells, probably causing more noticeable hyperprolactinemia induced clinical phenotype. Also we observed that four out of five PRL or GH secreting PitNETs were expressing more than one cell lineage marker while the same was true in two out of six PitNETs which were designated as non-functioning PitNET. Similar differences between secretion subtype and cell lineage markers has been observed in other studies [15] and they are of low concern regarding detection of somatic tumor variants."

6. Line 104: “We show that the sequencing data analysis of PitNET benefits from using multiple variant calling approaches”. Past tense should be used. Also, current evidence does not support the conclusion that “multiple variant calling approaches” benefit “data analysis of PitNET”.

Author answer: We have changed the wording to make the statement more precise: "We believe that the sequencing data analysis of PitNET benefits from using multiple variant calling approaches by providing higher confidence in variants that were detected with different methods".

Minor:

Line 27: change “Two samples” to “For each patient, two samples”.

Line 345: change “largePitNET” to “large PitNET”?

Author answer: Thank you this, indeed, improves clarity. We have edited accordingly.

 

Reviewer #2: The manuscript PONE-D-20-40370 entitled ‘Whole exome sequencing reveals novel risk genes of pituitary neuroendocrine tumors’ identify novel mutations in pituitary tumors. The study is really interesting from the clinical point of view, since it validates some of previously described mutations and provides evidence of new candidate genes. It is not a large cohort, but it is still informative work. Nevertheless, there are some issues that may be addressed.

Author answer: Thank you for reviewing and considering our research manuscript.

1. The pituitary tumor subtype is a key element of these kind of studies, and it must be indicated in the abstract and introduction. Although it may be clear in the figure, a more detailed characterization including IHC should be included, perhaps a non-gonadotroph but ACTH- or GH-silent adenomas were included, which may be interesting.

Author answer: We agree that IHC information was somewhat limited and didn't provide full overview about the sequenced samples. Therefore, we have added subsection in Results: "Immunohistochemistry staining results of PitNET samples" with a text: "Immunohistochemistry data of paraffin-embedded PitNET tissues were obtained for 11 PitNET patients using hormone antibodies (except CGA, where eight samples were analysed) and also 11 but different samples for lineage marker antibodies (S2). Concordance of hormone antibody immunohistochemistry with clinical diagnosis was observed in eight out of 11 patients, meanwhile nine out of 11 patients had agreement between clinical diagnosis and ICH data for PitNET lineage markers (PIT1, NR5A, TBX19). Most of NFPA clinical diagnoses actually were gonadotroph hormones expressing PitNETs. Few notable exceptions were observed: PN02 (with clinical NFPA diagnosis) was positive for GH and ACTH as antibody staining showed up to 30% of cells containing these hormones. PN04 (NFPA) had no detectable hormone expression, but PN05 (NFPA) was expressing every tested hormone, with high levels of GH, PRL and ACTH showing up in the data, although the highest expression level was for glycoprotein alpha subunit (required to produce gonadotropic hormones). PN10 (clinically PRL secreting PitNET) and PN14 (clinically GH secreting PitNET) were also expressing high levels of the glycoprotein alpha subunit.

PitNET cell lineage markers showed difference from clinical diagnosis in PN02 and PN10. PN02 is expressing corticotroph cell lineage marker in 30 – 70% of its cells. PN10 which clinically manifests as prolactin secreting PitNET is expressing higher amount of NR5A which is gonadotroph cell lineage marker than PIT1 associated with development of prolactin secreting PitNET. " And added paragraph in Discussion: "Regarding IHC results confirmed that most of the PitNET presented clinical phenotype according to their expressed lineage markers and hormones. Only one of the IHC tested clinically non-functional PitNET (PN02) differed from the rest by having TPIT cell lineage marker ekspressed in 30% – 70 % of observed cells rather than NR5A1. On the other hand, PN10 was presenting hyperprolactinemia phenotype, but the highest proportion of cells (>70%) was expressing CGA and NR5A1 indicating gonadotroph PitNET [28], nevertheless, PN10 was also expressing POU1F1 cell lineage marker and PRL in up to 30% of cells, probably causing more noticeable hyperprolactinemia induced clinical phenotype. Also we observed that four out of five PRL or GH secreting PitNETs were expressing more than one cell lineage marker while the same was true in two out of six PitNETs which were designated as non-functioning PitNET. Similar differences between secretion subtype and cell lineage markers has been observed in other studies [15] and they are of low concern regarding detection of somatic tumor variants."

2. Since there is a vast range of mutation per sample, from 0 to 31, is there any correlation with clinical or molecular characteristics?

Author answer: This indeed is interesting point, we did several correlation analyses between mutation count and available clinical characteristics and did not observe any significant correlation.

3. In the discussion, the authors indicate that the use of different workflows is advantageous, nevertheless, they do not really compare the methods, since they combine the results of the different workflows used.

Author answer: Thank you for this important point indeed we wanted to look at different algorithms that are using different methods to distinguish between true and false signal in the NGS data to understand more reliable data in our case. During the procedure of the analysis we have come to the conclusion that the somatic variants that are detected by one algorithm but not by other are not necessarily false positives, but more likely lack confidence in certain areas of variant detection. We decided to select variants with the highest confidence with the knowledge that they do not represent 100% of possible finds. We have included this information at the beginning of the discussion and provided references (such as 19 Miller et al.) where readers can explore this field further. Therefore, at the end of the analysis we reframed our aim not to compare the workflows but to obtain the most reliable somatic mutation data describing our PitNet samples.

 

Reviewer #3: This article reported results for the identification of two novel PitNET candidate genes ( AC002519.6 and AHNAK) with recurrent somatic variants in a PitNET cohort and found 13 more genes overlapping from previous PitNET studies containing somatic variants. This article does not show relevant and robust results in the field of PitNETs. Thus, there are major aspects that could be review aiming to improve the manuscript:

Author answer: Thank you for careful and constructive review of our manuscript and showing additional points how we could improve the study.

- The authors should considerably improve the result section. In general, the results are insufficient since the authors could acquire more results from the methodology used and data analysis. Then, we encouraged the author to deeply analyzed the data obtained from both methodologies used (GATK and Strelka2).

Author answer: Thank you for this important comment we have edited the results section to improve result representation, including description of immunohistochemstry results, description of main somatic variant findings, added S2 table legend, added and clarified data to Fig1. Indeed, we have considered to more deeply compare the results provided by both methodologies used (GATK and Strelka2) as different algorithms that are using different methods to distinguish between the true and false signal in the NGS data and to understand more reliable data in our study. During the data analysis we have come to the conclusion that the somatic variants that are detected by one algorithm but not by other are not necessarily false positives, but more likely lack confidence in certain areas of variant detection. To more properly represent the data of PitNET somatic variant we decided to select variants with the highest confidence with the knowledge that they do not represent 100% of possible finds. We have included this information at the beginning of the discussion and provided references (such as 19 Miller et al.) where readers can explore this field further. Therefore, at the end of the analysis we reframed our aim not to compare the workflows but to obtain the most reliable somatic mutation data describing our PitNet samples and concentrate on representing the somatic variation landscape of PitNETs.

- The sample cohort is only for 15 samples, being only three GH secreting and two PRL secreting tumors. Thus, the authors should complete the cohort with more samples for each subtype to avoid bias in the data analysis or reanalyzed other cohorts with the same algorithm in order to corroborate the data of your cohort.

Author answer: The recruitment of patients and sequencing of exomes has been completed in year 2018, as well as funding for this project. We have since moved to sequence genomes using different sequencing platform (MGI Sequencers) and therefore we will be unable to increase sample size using the same methods in all stages. Nevertheless, we think that our findings contribute positively towards field studying exomes of PitNETs and provide addional insight into genetic variation and makes such sequencing data available for possible future reasearch. We have identified recurring somatic variant in pseudogene (AC002519.6) and recurrent genes AHNAK, PALM2/AKAP2) in our sample. As well as potential candidates across PitNET studies (GNAS, TSC2, KLHL4, POLR3B, KHDRBS2, LAMA1, TUBGCP6, RYR1, PLA2G6, TESK1, AHNAK). Therefore, we think these data could serve as leads for future genetic and functional studies.

As a minor comment, I would like to suggest that the expression level in figure 1 should be represented as a log2(FC) and not as a percentage because they could be ambiguous.

Author answer: Thank you, here it is our problem with imprecise caption "Expression level". We have changed this caption to "% of cells expressing lineage marker" in legend and also replaced text in figure describing PitNET characteristics: "Expression level of PA lineage markers" with "% of cells expressing PitNET lineage marker"

In conclusion, this article needs to improve major/minor aspects to achieve an appropriate relevance.

Author answer: Thank you once again for insightful review and pointing out issues with the manuscript, we tried to take into account and join suggestions of all three reviewers.

---

## [Decision Letter · Decision Letter 1]

1 Mar 2022

Whole exome sequencing reveals novel risk genes of pituitary neuroendocrine tumors

PONE-D-20-40370R1

Dear Dr. Peculis,

We’re pleased to inform you that your manuscript has been judged scientifically suitable for publication and will be formally accepted for publication once it meets all outstanding technical requirements.

Kind regards,

Raul M. Luque, PhD

Academic Editor

PLOS ONE

Reviewers' comments:

Reviewer's Responses to Questions

**Comments to the Author**

1. If the authors have adequately addressed your comments raised in a previous round of review and you feel that this manuscript is now acceptable for publication, you may indicate that here to bypass the “Comments to the Author” section, enter your conflict of interest statement in the “Confidential to Editor” section, and submit your "Accept" recommendation.

Reviewer #2: All comments have been addressed

Reviewer #3: All comments have been addressed

2. Is the manuscript technically sound, and do the data support the conclusions?

Reviewer #2: Yes

Reviewer #3: Yes

3. Has the statistical analysis been performed appropriately and rigorously? 

Reviewer #2: Yes

Reviewer #3: Yes

4. Have the authors made all data underlying the findings in their manuscript fully available?

Reviewer #2: Yes

Reviewer #3: Yes

5. Is the manuscript presented in an intelligible fashion and written in standard English?

Reviewer #2: Yes

Reviewer #3: Yes

6. Review Comments to the Author

Reviewer #2: All comments have been addressed. No further comments.

Reviewer #3: (No Response)

7. PLOS authors have the option to publish the peer review history of their article (what does this mean?). If published, this will include your full peer review and any attached files.

Reviewer #2: No

Reviewer #3: No

---

## [Editor Report · Acceptance letter]

29 Mar 2022

PONE-D-20-40370R1 

Whole exome sequencing reveals novel risk genes of pituitary neuroendocrine tumors 

Dear Dr. Peculis:

I'm pleased to inform you that your manuscript has been deemed suitable for publication in PLOS ONE. Congratulations! Your manuscript is now with our production department. 

Kind regards, 

on behalf of

Dr Raul M. Luque 

Academic Editor

PLOS ONE